# The Management of Workplace Violence against Healthcare Workers: A Multidisciplinary Team for Total Worker Health^®^ Approach in a Hospital

**DOI:** 10.3390/ijerph20010196

**Published:** 2022-12-23

**Authors:** Reparata Rosa Di Prinzio, Giorgia Bondanini, Federica De Falco, Maria Rosaria Vinci, Vincenzo Camisa, Annapaola Santoro, Marcello De Santis, Massimiliano Raponi, Guendalina Dalmasso, Salvatore Zaffina

**Affiliations:** 1Alta Scuola di Economia e Management dei Sistemi Sanitari (ALTEMS), Università Cattolica del Sacro Cuore, 00168 Rome, Italy; 2Occupational Medicine Function Unit, Health Directorate, Bambino Gesù Children’s Hospital IRCCS, 00165 Rome, Italy; 3Health Directorate, Bambino Gesù Children’s Hospital IRCCS, 00165 Rome, Italy

**Keywords:** aggression, mental health, prevention, psychological well-being, health promotion, emotion, organization

## Abstract

The risk of aggression against healthcare workers (HCWs) is a globally well-known topic. However, workplace violence (WV) is often considered as part of HCW’s job, leading to a general underreporting. This cross-sectional study aims at providing a descriptive analysis of aggressive acts against HCWs registered in a 34-month period in a pediatric hospital. According to a specific protocol, each aggressive act was analyzed by a multidisciplinary team using the “Modified Overt Aggression Scale” (MOAS), the “General Health Questionnaire-12” (GHQ-12), and the “Short Form-36 Health Survey” (SF-36) to build a report addressing improvement measures. A three-domain model of WV was also developed considering: (1) assaulted HCWs, (2) attacker-related issues, and (3) environmental context. Contributing factors to overt aggression were outlined and tested using univariate analyses. Statistically significant factors were then included in a multiple linear regression model. A total of 82 aggressive acts were registered in the period. MOAS scores registered a mean value of 3.71 (SD: 4.09). Verbal abuse was the most common form of WV. HCWs professional category, minor psychiatric disorder, emotional role limitation, type of containment used, and emotion intensity were significantly associated with overt aggression (*p* < 0.05), as well as the attacker’s role in the hospital (*p* < 0.05). The multiple regression analysis confirmed these findings (*p* < 0.001). Raising awareness on the aggression risk and contributing factors may lead to a relevant improvement of workplace environment, individual workers’ health, and organizational well-being.

## 1. Introduction

Globally, violence is a widespread phenomenon in the workplace. Healthcare is one of the most affected sectors, and the most exposed categories are healthcare workers (HCWs), especially nurses and physicians [1]. American data indicate that about 70% of workplace aggressions occurred in health services and 10% of HCWs operating in the public sector have suffered violence-related consequences that led to absences from work, compared to 3% of HCWs in the private sector [2]. In European countries, 4% of the active HCWs population reported having experienced verbal and physical violence from people outside the workplace, such as patients or clients [3,4]. Data on WV greatly vary across Italian regions, although a high prevalence is recognized [5]. Evidence provided by the Italian Ministry of Health show that WV against HCWs represent almost 9% of the total adverse events reported throughout the national territory [2]. Psychological and verbal abuse such as screams or threats are the leading forms of workplace violence (WV), but physical attacks may also occur [1].

WV has notable negative effects on employees’ health, causing job work overload, decreased job satisfaction, fatigue, and exhaustion [6,7], which in turn often leads to high turnover and absenteeism rate [8]. Violent attacks can lead to serious experiences for HCWs, including psychological, physical, organizational, and professional consequences [9], and may also cause a reduction of the organization’s performance [10,11]. In the recent SARS-CoV-2 pandemic period, an increased workload caused high levels of frustration and distress, further burdening the HCWs [12].

Although aggressive events are broadly present, they are still underreported [13]. Consequently, HCWs are not prepared to face violence by colleagues and/or patients [1]. A passive acceptance by HCWs is usual, and aggressions are perceived as a regular component of their work [1]. In this study, we address this trend by reporting a description of data on aggressive acts among HCWs suggesting a model for the assessment of contributing factors related to the development of overt aggression.

## 2. Materials and Methods

### 2.1. The Hospital Protocol for the Management of Workplace Violence against HCWs (MWVaH Protocol)

Over the past four years, a specific protocol for the management of WV against HCWs (MWVaH protocol) has been implemented in the pediatric hospital. The procedure involves the reporting by the worker (or their supervisor) of the assault suffered using a digital platform for sentinel events that is independently accessible online by all hospital employees. Entry of the description of the assault event automatically generates a report that is analyzed by a multidisciplinary team of the Health Directory, which includes occupational physicians and a psychologist. The team prepares an ad hoc structured interview with the assaulted worker to investigate the matter further, using the root cause analysis approach [14]. Details are acquired regarding the form of aggression experienced (physical, verbal, against objects, and against self), factors that may have contributed, outcomes of the event, and containment interventions taken using the structured “Modified Overt Aggression Scale” (MOAS). The consequences to the health worker are focused on through the administration of a clinical questionnaire consisting of the “General Health Questionnaire-12” (GHQ-12) and “Short Form-36 Health Survey” (SF-36) scales investigating mental and general health, respectively. Critical issues found in the management of the episode are then identified and consequently improvement actions are finally proposed to minimize the risk of WV.

### 2.2. The Novel Theoretical Framework for the Management of Aggressions against HCWs

Several social and psychological models are available in the literature to understand the underlying mechanisms and origins of violence [15,16,17,18]. However, there are no models specifically contextualized in the field of work. Thanks to the application of the MWVaH protocol, it is possible to outline a new theoretical framework in healthcare facilities from the observation of the aggressive acts we recorded. In the scenario of aggressive acts, two main components were outlined: first, the conflictual relationships between the HCW and the aggressor; second, the specific environment in which the aggressions occurred represents a theatre that contributes to the escalation process, due to multiple factors that may play a triggering role for overt aggressions. In this study, after reporting a detailed descriptive analysis of aggressive acts, we investigated the role of factors contributing to overt aggression in relation to three domains: HCW’s characteristics, attacker’s characteristics, and environmental context. Therefore, we explored our theoretical framework by analysing the following data from the three domains:(1)Assaulted HCWs, regarding age, sex, professional category, role in the organization, mental health, general health, and objective and subjective factors;(2)Attacker-related issues, investigating the role of attackers in the hospital, and the characteristics of the hospitalization-related features;(3)Environmental context, concerning spatial and temporal distribution of aggressive episodes.

### 2.3. Study Design

A cross-sectional study has been set to analyze characteristics of the aggressive acts reported by HCWs in the hospital in a 34-month period from March 2019 to December 2021. Informed consent was obtained from all the participants.

The analysis of WV recorded in the hospital was conducted in two steps: (1) the evaluation of aggressive acts registered in the hospital according to the proposed model of WV in healthcare settings, and (2) the evaluation of contributing factors to overt aggression.

#### 2.3.1. The Description of Aggressive Acts

A detailed description of aggressive acts was performed using the “Modified Overt Aggression Scale” (MOAS) [19], Italian version [20], which provided the level of aggressive acts. The scale consists of four questions that investigate the four forms of aggression: verbal, e.g., “repeatedly or deliberately threatens violent actions against others or self”; against objects, e.g., “throws objects violently”; against oneself, e.g., “inflicts serious injury or attempts suicide”; and physical, e.g., “attacks others causing serious injury (fractures, broken teeth, deep cuts, loss of consciousness)”. These questions ranged on a 5-point Likert scale (0–4) from “no aggression manifested” to a worst-case scenario. Partial scores resulted using weighting factors for each question: (1) verbal aggression (scored from 0 to 4, weighting factor: 1); (2) aggression towards objects (scored from 0 to 8, weighting factor: 2); (3) self-inflicted violence (scored from 0 to 12, weighting factor: 3); and (4) physical aggression (scored from 0 to 16, weighting factor: 4). The total score is computed as the sum of partial scores ranging from 0 to 40; the higher the score the worse the aggressive act.

#### 2.3.2. First Domain: The Psychological Insight of the Assaulted HCWs

Age, sex, and occupational data regarding professional category (e.g., nurses, physicians, others) and role in the organization (e.g., managerial or executive) were outlined for the assaulted HCWs. The psychological insight of the assaulted HCWs was explored in terms of consequences that the aggression had on mental and general health. The “General Health Questionnaire-12” (GHQ-12) identifies the change in the normal psychic functioning towards personality disorders and the adaptation patterns associated with distress due to new stressful phenomena [21]. Each question is ranged on a 4-point Likert scale, with a total score ranging from 0 to 36 points; the higher the score the lower the mental health; we assumed scores over 21 as needing intervention.

The “Short Form-36 Health Survey” (SF-36) is a self-administered questionnaire aimed to quantify general health status and measure the health-related quality of life. The structure comprehends eight subscales regarding physical functioning (10 items), limitations due to physical health (4 items), physical pain (2 items), general health perception (5 items), vitality (4 items), social activities (2 items), limitations due to emotional problems (3 items), and emotional well-being (5 items). The last item differently assesses the change in health status compared to the previous year. Each item is ranged on a 5-point Likert scale, and each subscale is then scored on a 0–100 scale considering a weighted sum of items [22]. The higher the score, the better the perceived health; subjects with scores equal to or higher than 60 were considered to have good general health.

A factor analysis of the management of aggressive acts put in place by assaulted HCWs was performed considering two factors:Objective factor: type of containment enacted (e.g., verbal, physical, verbal and physical);Subjective factors (perceptions): immediate behavioral strategy intended to reduce possible consequences of the aggressive event and emotions felt (e.g., anger, frustration, fear, disappointment, sadness, and injustice) and their intensity at the time of the aggression.

#### 2.3.3. Second Domain: Attacker-Related Issues

For profiling attacker we considered two areas:The attacker’s role in the healthcare organization (e.g., caregiver, patient, colleague);The characteristics of the hospitalization, exploring age of the recovered patients, duration of hospitalization, and reason for admission (e.g., acute situation or relapse, chronic illness care); performed surgical operation.

#### 2.3.4. Third Domain: The Environmental Context

Spatial and temporal distribution of aggressive episodes were registered in terms of hospital setting where the aggression occurred and the time of day. Operative units were classified in four categories, regarding high-complexity/long-term units (e.g., intensive care units, child neuropsychiatry, neurorehabilitation), emergency admission units (e.g., emergency department), ordinary admission units (e.g., cardiology, surgery units), and outpatient units (e.g., dentistry, blood collection centre). Temporal distribution was considered according to work shift of the assaulted HCW into three categories (e.g., morning, afternoon, and night).

### 2.4. Statistical Analyses

The descriptive analysis of aggressive acts and demographics characteristics of the population of assaulted HCWs were set up using mean and standard deviation for continuous variables (e.g., assaulted HCW’s age, questionnaire’s scores, hospitalization length) and frequency for categorical variables (e.g., assaulted HCW’s sex, professional category, immediate behavioural choices and emotions, attacker’s role, age of the recovered patients, reasons for hospitalization, hospital setting, time of the day, and improvement actions suggested). The relationship between the three domains in the escalation process to overt aggression was independently evaluated using univariate analysis of variance, considering MOAS score as dependent variable and data on assaulted HCWs, attacker-related issues, and environmental context as independent variable. Then significant variables were included in a multiple linear regression model.

## 3. Results

### 3.1. Descriptive Analyses of Aggressive Acts Registered in the Hospital

#### 3.1.1. The Form of Aggressive Acts and Their Evaluation

A total of 82 aggressions have been registered in the period. In 58.5% of aggressive episodes more than one HCW was involved. MOAS scores registered a mean value of 3.71 (±4.09). Among the four forms of violence, verbal abuse was the most common. The occurrence and correspondent MOAS scores of the four forms of WV is shown in Table 1.

#### 3.1.2. The Assaulted HCWs’ Perspective in WV Management

Overall, assaulted HCWs were aged 44.77 ± 10.09; they were mainly females (n = 68, 82.9%). They were mainly nurses (n = 56, 68.3%), followed by physicians (n = 20, 24.4%); other professional categories including technicians, socio-sanitary workers, and administrative personnel accounted for the 7.2% (n = 6). HCWs had mostly an executive role (n = 71, 86.6%).

According to the GHQ-12 scores, 15 assaulted HCWs (18.3%) had a psychological impairment requiring intervention; a psychological support at the workplace was chosen in 5 of them. Insufficient general health was registered in 16 subjects (19.5%). Mean scores of the administered questionnaires are shown in Table 2.

According to the MWVaH protocol, all adverse events were reported to the Health Directory within the next 24 h. Moreover, a report of injury at work was made to the competent National Institute for Insurance against Accidents at Work in two cases and in one case a legal complaint was executed. Overall, only one HCW reported an effective management of the event.

Objective factorial analysis of the management of aggressive episodes put in place by assaulted HCWs showed that verbal actions were sufficient in most of the situations (92.9%), whilst physical containment was necessary in four cases, and both verbal/physical containments were implemented in two cases. Subjective factor analysis of the management showed that immediate behavioural strategy consisted in the early detection of the attacker (70.7%), followed by the communication techniques (24.4%), and the efficient surveillance system (15.9%). HCWs reported to have experienced strong emotions at the time of the aggression. Anger and frustration were the most referred to (47.6%), as well as fear (36.6%); other perceptions regarded disappointment (14.6%), sadness, and injustice (3.7%) (Table 3).

#### 3.1.3. The Attacker-Related Issues

In 71 cases, the caregiver was responsible for the aggressive act (86.4%), whereas patients were involved in 7 events (8.6%); peer aggressions occurred in 4 cases (4.9%). Factors related to the hospitalization showed that hospitalized patients were 9 years of age on average (SD: 5.9 years) and the mean duration of hospitalization was 54.79 days (SD: 110.84 days). Reasons for admission mostly regarded acute situations (67.1%), while the residual part were due to chronic illnesses (32.9%). In 17.1% of cases, a surgical operation was performed during the hospitalization period (Table 4).

#### 3.1.4. The Environmental Context of Registered WV

Aggressions generally occurred during day shifts (72.2%). High-complexity and long-term units were the most involved settings (48.8%), followed by ordinary admission units (23.2%) and emergency admission units; outpatient units were less involved (11.0%) (Table 5).

### 3.2. The Evaluation of Contributing Factors to Overt Aggression

Univariate analyses evidenced that professional category, the presence of minor psychiatric disorder, emotional role limitation, type of containment used to counteract the aggressive act, and emotion intensity were HCWs factors which contribute to overt aggression. Attacker’s role was also a contributing factor, whereas no environmental factors can predict overt aggression (Table 6).

## 4. Discussion

Aggression is a universal behavioral trait among animals used as a mechanism to establish power over someone or to defend from a perceived threat [23]. Aggressive acts are a visible expression of a misalignment of the relationship between individual characteristics and the environmental context. In healthcare settings, the essential trigger lives in the relationship between healthcare personnel and users. Our findings showed that the escalation process to overt aggression is influenced by the two dimensions of this human relationship, regarding on the one hand HCWs’ professional category, psychological well-being, perception of emotion intensity and consequent emotional role limitation, and the user’s role in the hospital.

The impact of the emotional role in health professional categories can generate symptoms of exhaustion by lowering the level of employees’ health [6], thus starting a vicious cycle which mines the organizational climate. Horizontal aggression can increase too based on the level of dissatisfaction, work overload, fatigue, and burnout [7]. Furthermore, consequences may be experienced at a behavioral level (e.g., laziness), emotional level (e.g., anger, fear), psychological level (e.g., mood disorders, burnout), and physical level (e.g., headache, heart pounding) [24,25,26]. Although there is a major prevalence of female workers in the health sector, our results confirm a gender difference related to WV, with females having a higher risk than males [2,27].

A peculiar feature of the pediatric world concerns the relationship between the patient’s family and HCWs. This figure is connected to the wear and tear of the caregiver: the complexity of the patient and the length of hospitalization are critical elements to be considered. WV triggering factors include prolonged waiting time and inadequate communication as well as organizational and ergonomic criticalities [28]. Relationships experienced by HCWs with patients and their caregivers (e.g., family members) may have a significant impact on aggressions [28]. Our results confirmed that verbal aggressions are more common than physical aggression [2,29,30].

Several occupational factors may influence the occurrence of violen acts in the workplace, such as organizational and structural issues regarding miscommunication and poor environmental design [31]. In addition, night shifts were found to be a risk factor of aggression, due to poor lighting and limited visibility [32]; conversely, in our sample population for the most part of aggression occurred during day shifts. Although the emergency department notably face a high prevalence of WV [33,34], a general underreporting is confirmed in our sample [35]. Moreover, our evidence shed light on the great frequency of aggressive acts in high-complexity units, which have a higher risk of WV due to the presence of organizational multispecificity in our pediatric hospital. This phenomenon is probably caused by the wear and tear on the family caregiver of patients suffering from chronic illnesses. In fact, the caregiver burden may result from the constant assistance required by chronic patients in need of complex treatment [36].

Despite the limited sample, which is the main limitation of our study, our domain analysis may help in understanding the etiology of aggressive acts, which can aid HCWs in the prevention and management of WV. Further studies are needed to gather systematic evidence of this phenomenon. However, the assessment of risk factors, addressing underreporting of violent episodes, and implementing WV management initiatives are successful organizational mitigation strategies, as suggested in the literature [37]. Workplace health promotion programs focused on a participative approach and employee engagement (e.g., psychological support [38], relaxation techniques, and yoga/mindfulness courses [39]) in healthcare may prevent work-related stress and preserve HCWs from the development of physical and mental fatigue [40,41,42].

In recent years, a growing interest in preventing psychosocial risks has been observed, specifically regarding the development of strategies to increase protective factors related to mutual social support networks and the development of coping skills [31]. Many aspects have been addressed to tackle WV in healthcare settings with the aim of improving the quality and safety of care and helping with clinical risk management. The development of targeted company policy for the prevention of WV, safety training programs for WV management, and courses on communication techniques for early recognition of potential aggressive and violent behaviors have been produced, as well as procedures for reporting and procedures to activate medical, psychological, and legal support after an episode of violence [2,29,30].

## 5. Conclusions

In recent years, a growing interest in preventing psychosocial risks has been observed regarding the development of strategies to increase protective factors related to mutual social support networks and the development of coping skills [31]. In the hospital we addressed many aspects to tackle WV in healthcare settings with the aim of improving the quality and safety of care and helping with clinical risk management. The development of targeted company policy for the prevention of WV, safety training programs for WV management, and courses on communication techniques for early recognition of potential aggressive and violent behaviors have been produced, as well as procedures for reporting and procedures to activate medical, psychological, and legal support after an episode of violence [2,29,30].

## Figures and Tables

**Table 1 ijerph-20-00196-t001:** Occurrence and MOAS scores of the four forms of WV (in order of frequency).

Form of Aggression	Frequency n (%)	Mean ± SD	Range
Verbal aggression	76 (92.7)	1.91 ± 1.09	0–4
Physical aggression	21 (25.6)	1.54 ± 3.07	0–12
Aggression against objects	12 (14.6)	0.46 ± 1.19	0–4
Aggression against self	2 (2.4)	0.17 ± 1.01	0–6
MOAS total score (points)	82 (100.0)	3.71 ± 4.09	0–26

Notes: MOAS: Modified Overt Aggression Scale; SD: standard deviation.

**Table 2 ijerph-20-00196-t002:** Questionnaire scores on mental and general health.

Scale	Range	Mean ± SD
GHQ-12	1–26	12.15 ± 7.10
SF-36 total score	31–97	76.25 ± 17.84
subscale 1—physical activity	35–100	91.35 ± 15.00
subscale 2—physical role limitation	33–100	84.92 ± 21.64
subscale 3—physical pain	32–100	79.81 ± 24.31
subscale 4—general health	40–97	70.73 ± 14.61
subscale 5—vitality	15–100	58.46 ± 21.95
subscale 6—social activities	0–100	72.35 ± 27.90
subscale 7—emotional role limitation	0–100	83.19 ± 28.81
subscale 8—mental health	16–100	69.23 ± 21.25

Notes: GHQ-12: General Health Questionnaire-12; SD: standard deviation; SF-36: Short Form-36 Health Survey.

**Table 3 ijerph-20-00196-t003:** Factor analysis of WV management by the assaulted HCWs.

Factor			Frequency n (%)	Mean ± SD
Objective factor	Type of containment	Verbal	76 (92.9)	-
		Physical	4 (4.3)	-
		Verbal and physical	2 (2.8)	-
Subjective factors	Immediate behavioral actions	Early detection of the attacker	58 (70.7)	-
		Communication techniques	20 (24.4)	-
		Surveillance system	13 (15.9)	-
		Securing of the attacker	9 (11.0)	-
		Presence of colleagues	8 (9.8)	-
		Presence of superiors	3 (3.7)	-
	Emotions	Anger	39 (47.6)	-
		Frustration	39 (47.6)	-
		Fear	30 (36.6)	-
		Disappointment	12 (14.6)	-
		Sadness	3 (3.7)	-
		Injustice	3 (3.7)	-
		Intensity of any emotion (0–5)	82 (100.0)	3.6 ± 1.5

Note: SD: standard deviation.

**Table 4 ijerph-20-00196-t004:** The attacker’s role and hospitalization factors related to aggressive acts.

Domain		Frequency n (%)	Mean ± SD
Attacker’s role	Caregiver	71 (86.6)	-
	Patient	7 (8.5)	
	Colleague	4 (4.9)	
Hospitalization factors	Patient’s age (years)	-	9.0 ± 5.9
	Duration of hospitalization (days)	-	54.8 ± 110.8
	Reason for admission		
	Acute illness	55 (67.1)	-
	Chronic illness	27 (32.9)	-
	Surgical operation	14 (17.1)	-

Note: SD: standard deviation.

**Table 5 ijerph-20-00196-t005:** Spatial and temporal distribution of aggressive acts.

Distribution		Frequency n (%)
Spatial distribution	High-complexity/Long-term units	40 (48.8)
	Emergency admission units	14 (17.1)
	Ordinary admission units	19 (23.2)
	Outpatient units	9 (10.9)
Temporal distribution	Morning	30 (36.6)
	Afternoon	29 (35.4)
	Night	22 (26.8)

**Table 6 ijerph-20-00196-t006:** Univariate analyses and multivariate regression model.

Domain	Variable	B	Adjusted R^2^	*p* Value
Assaulted HCWs	Age	0.191	0.037	0.115
	Sex	−0.079	0.008	0.516
	Professional category	0.255	0.051	0.033
	Role in the organization	−0.099	−0.005	0.414
	GHQ-12	0.461	0.175	0.027
	SF-36	−0.296	0.042	0.181
	subscale 1—physical activity	0.079	−0.043	0.725
	subscale 2—physical role limitation	−0.345	0.075	0.116
	subscale 3—physical pain	−0.228	0.004	0.308
	subscale 4—general health	0.149	−0.027	0.508
	subscale 5—vitality	−0.220	0.001	0.325
	subscale 6—social activities	−0.328	0.063	0.136
	subscale 7—emotional role limitation	−0.479	0.191	0.024
	subscale 8—mental health	−0.294	0.041	0.184
	Type of containment	0.569	0.323	<0.001
	Immediate behavioral strategy			
	early detection of the attacker	−0.109	0.012	0.389
	communication techniques	−0.079	0.006	0.532
	surveillance system	−0.153	0.023	0.224
	securing of the attacker	−0.131	0.017	0.299
	presence of colleagues	0.035	0.001	0.780
	presence of superiors	−0.095	0.009	0.450
	Emotion intensity	0.431	0.186	0.040
Attacker-related issues	Patient’s age	0.308	0.095	0.092
	Duration of hospitalization	−0.066	0.004	0.720
	Reason for admission	−0.154	0.024	0.417
Environmental context	Spatial distribution	−0.127	0.016	0.294
	Temporal distribution	0.064	0.004	0.598
Multivariate Regression Model		0.870	0.756	<0.001

## Data Availability

The data presented in this study are available on request from the corresponding author. The data are not publicly available due to ethical restrictions.

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
