# Peer review of "The Management of Workplace Violence against Healthcare Workers: A Multidisciplinary Team for Total Worker Health® Approach in a Hospital"

_ijerph, 2022, doi:10.3390/ijerph20010196_

Round 1
Reviewer 1 Report
This study is interesting, but it needs to be modified before being accepted.
1.The sum of some percentages in the table is not 100%, which needs to be checked.
2.p=0.000 need to change to"p<0.001"
Author Response
Dear Reviewer,
thank you for your observations and suggestions. We have provided point-by-point response in the attachment.
Kindest regards,
Reparata Rosa Di Prinzio, corresponding Author

Reviewer 2 Report
Since workplace violence is often considered as part of healthcare works' job leading to a general underreporting, this study aims at providing a descriptive analysis of aggressive acts against healthcare works registered in a 34-month period in a pediatric hospital.
Methodologically, according to a specific protocol, each aggressive act was analyzed by a multidisciplinary team using the “Modified Overt Aggression Scale”, the “General Health Questionnaire-12” (GHQ-12) and the “Short Form-36 health survey” (SF-36) used to build up a report addressing improvement measures. A three-domain model was also developed, with factors contributing to overt aggression outlined and tested using univariate analyses.
In the end of the paper, statistically significant factors were then included in a multiple linear regression model, with some conclusions and implications proposed.
Overall, the topic discussed in the paper is important, and I encourage the authors to deepen the research. I also suggest the authors to revise the paper.
First, the research should be improved by highlighting the theoretical contributions. Much works have been done on description of phenomenon, but a novel theoretical framework should be developed.
Second, the structure of the paper should be modified according the format of an academic paper, For example, some hypotheses are expected to be presented based on the theoretical framework.
Author Response

(The authors gave the same response as above.)

Reviewer 3 Report
Thank you for the opportunity to review the manuscript titled The Management of Workplace Violence Against Healthcare Workers: A Multidisciplinary Approach in an Italian Hospital. This paper investigates multiple variables associated with hospital workplace violence and seeks to understand contributing factors that are associated with increased violence.
The manuscript is fairly well written, with some peculiar and extraneous phrases that disrupt the overall flow and understanding of the paper. However, my recommendations for improvement for this manuscript have more to do with style than scientific rigor.
Abstract: Clear and concise. Line 16 requires a comma after the words "part of HCW's job"
Introduction:
Line 40: This sentence is awkward, and the words "occurred in services health or social issues" might be intended to read "health and social services" perhaps?
Line 50: This paragraph should begin "WV has notable..."
Line 51: The word "lead" should be changed to "leads"
Line 56: It is unclear what "distress grieving" means. Perhaps drop the word grieving?
Lines 94-105: This is a discussion about the Modified Overt Aggression Scale. I am not familiar with this scale, and the authors describe the Likert Scale anchors as ranging from 0-4, with the lowest anchor meaning" no aggression manifested" and the highest meaning "repeatedly or deliberately threatens violent actions against others or himself so as to obtain money or sexual services." If this is the actual anchor, it seems that this scale is unsuitable for use with hospital workplace violence, in that I believe most violence erupts from anxiety or fear associated with diagnosis and treatment. If this is not the anchor, this paragraph needs to be re-written to reflect that this is an example of a situation being rated on the scale, and not part of the scale itself.
Additionally, the authors should do a better job of explaining how the four types of violence (i.e., verbal, against objects, against self, and physical) are distinct from one another. For example, would throwing an object at the wall be considered against objects or physical?
Line 133: I am not clear on what the words "otherwise insufficient" have to do with this sentence. Perhaps they should be removed, or at least reworded.
Lines 170-171: Reword this sentence to begin "Then significant variables were included..."
Results:
Lines 222-223: Reword this sentence to read "Aggressions generally occurred during day shifts..." Also, the word "interested" is confusing. Perhaps use the word "involved"?
Discussion:
Lines 244-245: Omit the words "on the other hand"
Lines 246-256: This is the first mention of Diegrist's model of work-related stress. Was this used in the development of this study? If this model is germane to the paper, it should have been introduced much earlier in the Introduction, and then the discussion can be used to describe how the results reflect what was anticipated by the model.
Line 277: The use of the words "poor sample" is probably not what the author's intended to convey. Is their intent a "limited sample" given a small N or single setting?
Conclusions:
In my opinion, this section of the manuscript requires the most attention. The paper discusses several variables associated with the severity and antecedents of workplace violence in hospitals, but it is not an investigation on how to mitigate such violence. However, the conclusions section reads as if this was the focus and outcome of the paper. If the authors wish to link their study to interventions, which would be appropriate, this should be articulated as "future directions" for their research, and how a better understanding of workplace violence in hospitals can assist in intervention design.
Author Response

(The authors gave the same response as above.)

Round 2
Reviewer 2 Report
The authors have revised the manuscript according to the comments/suggestions. And I think it can be published.